# Pitfalls of Early Systemic Corticosteroids Home Therapy in Older Patients with COVID-19 Pneumonia

**DOI:** 10.3390/geriatrics7010021

**Published:** 2022-02-17

**Authors:** Chukwuma Okoye, Sara Rogani, Riccardo Franchi, Igino Maria Pompilii, Alessia Maria Calabrese, Tessa Mazzarone, Elena Bianchi, Bianca Lemmi, Valeria Calsolaro, Fabio Monzani

**Affiliations:** Geriatrics Unit, Department of Clinical & Experimental Medicine, University Hospital of Pisa, 56126 Pisa, Italy; chukwuma.okoye@phd.unipi.it (C.O.); sara_rogani@hotmail.it (S.R.); riccardo.franchi@me.com (R.F.); igino.pompilii@gmail.com (I.M.P.); alessiamariacalabrese@gmail.com (A.M.C.); tessa.mazzarone@gmail.com (T.M.); monnalisa.smile@hotmail.it (E.B.); bianca.lemmi@gmail.com (B.L.); fabio.monzani@med.unipi.it (F.M.)

**Keywords:** corticosteroids, older people, COVID-19, pneumonia, home therapy

## Abstract

Corticosteroids have been widely used for acute respiratory distress syndrome (ARDS), but their role in the early phase of SARS-CoV-2 infection is controversial. Our study aimed to determine the effectiveness of early corticosteroid therapy (ECT) in preventing the progression of disease, reducing the escalation of care and improving clinical outcome in older patients hospitalized for COVID-19 pneumonia. A total of 90 subjects (47.7% women; mean age = 82.3 ± 6.7 years) were enrolled. ECT was administered to 33 out of 90 patients before the hospitalization. At admission, no difference was detected in median SOFA score (2, IQR:2 vs. 2, IQR: 2). We found a significant difference in mean PaO_2_/FiO_2_ ratio during the first week of hospitalization between ECT patients and controls (F = 5.49, *p* = 0.002) and in mean PaO_2_/FiO_2_ ratio over time (F = 6.94, *p* < 0.0001). We detected no-significant differences in terms of in-hospital mortality and transfer to ICU between ECT patients and controls (27.1% vs. 22.8%, respectively, *p* = 0.63). ECT was associated with worse clinical outcomes, showing no benefit in attenuating the progression of the disease or reducing the escalation of care. These findings are crucial given the current pandemic, and further studies are needed to provide additional data on the optimal timing of initiating corticosteroid treatment.

## 1. Introduction

Severe Acute Respiratory Syndrome Coronavirus 2 (SARS-CoV-2) emerged in late December 2019 and spread globally at a formidable speed over the subsequent years, affecting tens of millions of people and causing more than one million deaths, leading the World Health Organization to declare a pandemic in March 2020 [1,2]. It is well-established that SARS-CoV-2 infection recognizes roughly three different stages: stage I, asymptomatic incubation; stage II, non-severe symptomatic period with the presence of virus; stage III, severe respiratory symptomatic stage with viral load. The focus on inflammatory status in patients with COVID-19 is built upon the knowledge that severe COVID-19 is due to immunological dysfunctions, including an impaired type I interferon response, increased inflammation, complement activation and endothelial stress [3].

Corticosteroids have been widely used in critically ill patients with acute respiratory distress syndrome (ARDS) and during the outbreaks of severe acute respiratory syndrome (SARS)-CoV and Middle East respiratory syndrome (MERS)-CoV to suppress the pro-inflammatory response and control the cytokine storm [4,5]. Recommendations on the use of corticosteroids for COVID-19 are largely based on data from the RECOVERY trial [6], where mortality at 28 day was lower among patients receiving corticosteroid compared with placebo. However, no benefit of dexamethasone was seen in patients who did not require supplemental oxygen at enrolment; notwithstanding, the use of glucocorticoids for COVID-19 early-phase pneumonia remains controversial, especially in older patients [6,7]. The aim of this prospective observational study was to assess the effectiveness of early corticosteroid therapy in preventing the progression of disease, reducing escalation of care and improving clinical outcomes in older patients admitted to an acute-care geriatric ward for COVID-19 pneumonia.

## 2. Materials and Methods

### 2.1. Study Design and Participants

All patients aged 65 or older hospitalized for COVID-19 pneumonia from May 2020 to March 2021 at the acute Geriatrics Unit of our University Hospital were consecutively enrolled. Clinical characteristics of the study population are reported in Table 1. At ward admission, the presence of positive history of chronic obstructive pulmonary disease, chronic heart failure, chronic obstructive pulmonary disease, arterial hypertension, diabetes mellitus, and dementia were recorded. The Charlson Comorbidity Index was also calculated [8]. Medication name, dosage and duration of home daily therapy was collected from all the patients at the admission interview or, in case of patients with dementia or without capacity, was asked to a caregiver by phone interview. Whenever possible, caregivers were requested to bring their medicine packets to the hospital, to confirm the number of pills previously taken. We categorized patients with early corticosteroid therapy (ECT) as those receiving oral corticosteroids home treatment for 2 days or longer, prior to hospital admission. All the patients underwent clinical examination and blood testing. The severity of respiratory failure was assessed by calculating the PaO_2_/FIO_2_ ratio (i.e., partial pressure arterial oxygen/fraction of inspired oxygen ratio) at Day 1–3–5–7 of hospital stay, respectively. A CT scan was performed within 48 h from the admission as part of the evaluation route for COVID-19 in the Emergency Department. According to hospital treatment guidelines for COVID-19, in case of severe cases of COVID-19 patients received the same protocol treatment, consisting of 4 to 6 mg dexamethasone daily, low-molecular-weight-heparin at prophylactic dosage and remdesivir (in the case of symptoms occurrence <10 days). Intra-hospital mortality or transfer to ICU were used as composite outcome. Written informed consent was obtained from all the patients; the legally authorized delegate provided informed consent in the case of patients temporarily or permanently without capacity. The study protocol complied with the Declaration of Helsinki and was approved by the Pisa University Hospital Ethic Committee.

### 2.2. Statistical Analysis

Statistical analysis was performed with R statistical software package (version 4.1.0, 18 May 2021 R Foundation for Statistical Computing, Vienna, Austria) and GraphPad Prism 9 (GraphPad Software, San Diego, CA, USA) was used to plot graphs. Continuous variables were presented as mean and standard deviation, ordinal variables as median and interquartile range (IQR), and categorical variables as percentage. Mann–Whitney and chi-square tests were used for multiple comparisons. In order to calculate the relationship between ECS dosage and respiratory failure, a Spearman’s correlation matrix was calculated between PaO_2_/FiO_2_ ratio and total oral corticosteroid received prior admission. In the case of daily assumption of dexamethasone and methylprednisolone tabs, we performed a prednisone-dose-equivalency conversion formula, as reported in a previous study [9]. A two-factor ANOVA for repeated measures was performed in order to evaluate the difference in means between patients receiving ECT and counterparts over the first seven days of hospitalization. A composite endpoint including in-hospital death or transfer to ICU due to clinical deterioration was also assessed. Tests were performed considering a level of significance of 5%.

## 3. Results

A total of 90 subjects (47.7% women; mean age 82.3, ±6.7 years) were evaluated in the study. Early corticosteroid therapy was administered to 33 out of 90 patients before hospitalization; median time of home-therapy assumption was 4 days (range 2–7). Overall, 29 patients received home-treatment with methylprednisolone, 3 patients received dexamethasone, and 1 patient received prednisone, as reported in Table 2. At hospital admission, compared to controls, patients with ECT experienced shortness of breath (57.5% vs. 43.8%, *p* = 0.21), and cough (30.3% vs. 21.3%, *p* = 0.32) more frequently; no differences were detected in median SOFA score (2, IQR:2 vs. 2, IQR: 2). Conversely, fever at admission to the ED was more frequent in patients not receiving ECT (47.3% vs. 45.5%, *p* = 0.86). Patients receiving early corticosteroid presented decreased PaO_2_/FiO_2_ ratio (median 261 (IQR 138) vs. 304 (IQR 106), *p* = 0.06) in comparison to the controls. Moreover, patients with ECT showed an increased white blood cells count (WBC), and lymphocytopenia (WBC mean 9254 ± 2561/mm^3^ vs. 6992 ± 3915/mm^3^, *p* < 0.01; lymphocytes mean 851 ± 579/mm^3^ vs. 1477 ± 2890/mm^3^, *p* = 0.32) than their counterparts. We detected higher baseline levels of C-reactive protein (mean 11.3 ± 7 vs. 6.1 ± 5.1 mg/dL, *p* = 0.002) in patients with ECT. Comparing chest CT findings, ECT patients had a higher prevalence of bilateral patchy shadowing (69.5% vs. 32.3%, respectively, *p* = 0.005), a lower proportion of pleural effusion (13% vs. 40.4%, *p* = 0.024). No significant differences were detected in terms of ground glass opacities (82.6% vs. 67.6%, *p* = 0.2) and pulmonary consolidations (43.3% vs. 48.6%, *p* = 0.69). Notably, patients with ECT had a significant worsening of PaO_2_/FiO_2_ ratio during the hospitalization (nadir median 138 (IQR 128) vs. 252 (IQR 191), *p* = 0.017). As shown in Figure 1, using the two-factor ANOVA for repeated measures, we found a significant difference regarding the mean of the PaO_2_/FiO_2_ ratio during the first week of hospitalization between patients receiving early ECT therapy and controls (F = 5.49, *p* = 0.002), and a statistically significant difference in mean PaO_2_/FiO_2_ ratio over time (F = 6.94, *p* < 0.0001). Furthermore, as shown in Figure 2, Spearman correlation test demonstrated an inverse correlation with a trend towards significance between ECT dosage and PaO_2_/FiO_2_ ratio at nadir (Rho = −0.248, *p* = 0.078). Finally, we did not find significant differences regarding the occurrence of the composite outcome (intra-hospital death or transfer to ICU) between ECT and controls (27.1% vs. 22.8%, respectively; *p* = 0.63).

## 4. Discussion

In our cohort of older patients hospitalized for COVID-19 pneumonia, those receiving early systemic corticosteroid home treatment showed a higher degree of respiratory failure during hospital stay compared with controls; moreover, we found non-significant differences in terms of in-hospital mortality. At hospital admission, patients receiving early ECT showed higher levels of inflammation markers along with an increased proportion of pulmonary interstitial injuries at chest CT. Despite the widely recognized immune-modulating beneficial effect of steroid therapy in COVID-19, timing and dosage of systemic corticosteroids remain pivotal for a favourable outcome, especially in immunocompromised patients [10].

The use of systemic corticosteroids in patients with COVID-19 has increased since the RECOVERY [6] trial demonstrated a 28-day mortality reduction in hospitalized patients with acute respiratory failure requiring oxygen therapy, although its ineffectiveness in the pre-symptomatic stage of the disease or in patients not requiring oxygen supplementation was also shown. On the other hand, a recent observational study reported that the initiation of corticosteroids within 72 h from hospital admission offered a significant mortality benefit. To date, few studies have attempted to investigate the effect of an early, at-home initiation of corticosteroids in oldest-old patients. Indeed, in the study by Bahl et al. the mean age of participants was 63.3 ± 14.5 years, similar to that of the RECOVERY trial (66.1 ± 15.7), [11,12]. This is a crucial factor since immune response is dynamically remodelled with advancing age, in a phenomenon called immune senescence. This condition increases susceptibility to infections, autoimmune disorders, and malignancies [13]; therefore, older patients are more prone to an altered activation of innate immune response, leading to a delayed resolution of inflammation and increased tissue damage [14].

In the current study, we found a relatively high prevalence of early corticosteroids home prescription, with a median time of ECT initiation of 4 days, and a 24-h median time of administration from symptom onset. At admission, patients receiving ECT presented a more severe degree of respiratory failure; interestingly, this gap increased during hospitalization, with patients in ECT having an almost 50% reduced PaO_2_/FiO_2_ at nadir compared to controls. Furthermore, we demonstrated a difference in the trend of mean PaO_2_/FiO_2_ ratio during the first seven days of hospitalization between patients given ECT and their counterparts, with the latter having a lesser degree of respiratory failure. Remarkably, the differences in mean PaO_2_/FiO_2_ increased over time, driving the test for interaction to approach statistical significance. Such a difference regarding the trend of respiratory failure could be explained by the reduced immune response related to an untimely assumption of corticosteroids, favouring viral replication due to its immunosuppressive action [15,16], and worsening the cytopathic damage to alveolar epithelial cells [16], especially in the initial phase of the infection. Furthermore, we found an inverse correlation between corticosteroids dosage and degree of respiratory impairment expressed by PaO_2_/FiO_2_ ratio; in other words, the higher the ECT daily dosage, the lower PaO_2_/FiO_2_ ratio found at hospital admission. Accordingly, patients receiving ECT presented higher baseline levels of C-reactive protein, with an increased prevalence of ground glass opacities and bilateral patchy shadows at chest CT than the controls. Although not statistically significant, we found a 40% higher in-hospital mortality compared to controls, probably due to the small sample size. These findings are in accordance with the study by Liu [17], that reported how an early administration of corticosteroids in severe COVID-19-related ARDS was associated with delayed viral clearance and an increased risk of death at 28 days and during hospitalization, especially in patients with early initiation and a high dose of corticosteroid treatment.

This combination of findings provides some support for the conceptual premise that an early corticosteroids administration during the first phase of COVID-19 is not beneficial in older patients; conversely, it is associated with a worse degree of respiratory failure, longer hospital stays and an increased risk of poor outcomes. However, with a small sample size, caution must be applied, as the findings might not be transferable to all severe COVID-19 older patients—we might have encountered a selection bias by comparing patients with different stages of COVID-19 with ECS patients that had longer days of symptoms before hospital admission. Nonetheless, the two cohort results were comparable, since all the patients showed severe COVID-19 pneumonia, and no significant differences were found in terms of amount of comorbidities, degree of dependence, or prevalence of dementia. Moreover, all the patients underwent the same hospital protocol for severe COVID-19 treatment, thus confirming the reliability of our findings.

## 5. Conclusions

Early corticosteroid home therapy showed no benefit in attenuating the progression or reducing the escalation of care in older patients with COVID-19 pneumonia. The detrimental effect might be related to the corticosteroid dosage as well as the illness severity. Indeed, patients with moderate-to-severe pneumonia might benefit from corticosteroid treatment in the second phase of the COVID-19 course, although this could have harmful results at the onset of the disease. Additional study into the optimal timing of initiating corticosteroid treatment, and to identify patients who could benefit most from corticosteroid therapy, is warranted. These findings are crucial given the current pandemic, and further, larger, multi-centre studies are needed to address these answers.

## Figures and Tables

**Figure 1 geriatrics-07-00021-f001:**
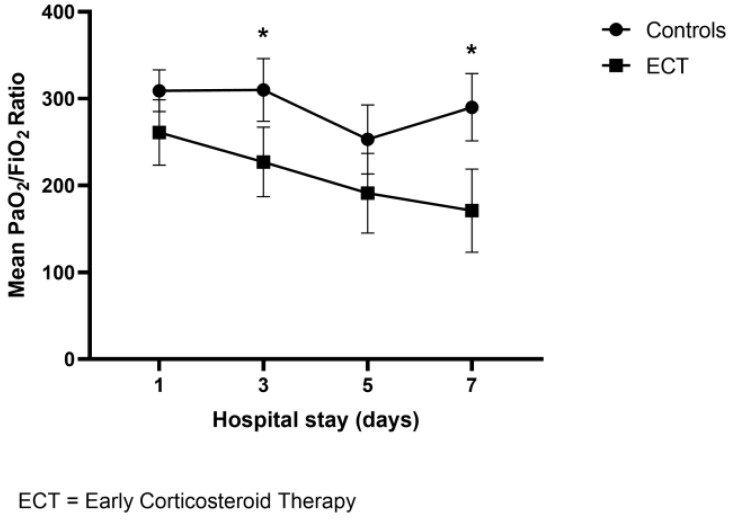
Trends of median PaO_2_/FiO_2_ ratio from hospital admission to Day 7. * *p* < 0.05.

**Figure 2 geriatrics-07-00021-f002:**
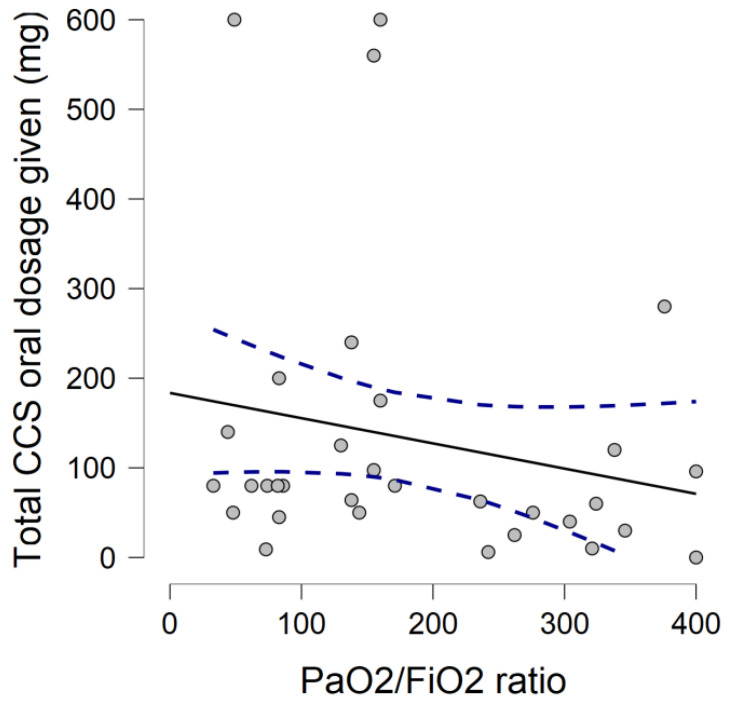
Correlation plot between PaO_2_/FiO_2_ ratio at nadir and total corticosteroid dosage received prior to hospital admission.

**Table 1 geriatrics-07-00021-t001:** Clinical characteristics of the study population.

	All Patients N = 90	ECT N = 33	Controls N = 57	*p*-Value
Gender (male)	47 (52.3)	16 (48.5)	31 (54.4)	0.58
Age (years)	82.3 (6.7)	84.5 (2.4)	81.3 (6.1)	0.02
SOFA score	2 (2)	2 (2)	2 (2)	0.6
Arterial Hypertension	58 (64.4)	15 (42.8)	33 (57.1)	0.18
Diabetes Mellitus	19 (21.5)	3 (9.0)	16 (29.2)	0.05
Chronic Heart Failure	32 (35.0)	12 (34.9)	20 (35.8)	0.92
COPD	33 (36.9)	12 (37.5)	21 (36.5)	0.94
Dementia	33 (36.9)	12 (37.5)	21 (36.5)	0.94
Charlson Comorbidity Index	5 (2)	5 (3)	5 (2)	0.45
Shortness of breath	44 (49)	19 (57.5)	25 (43.8)	0.21
Cough	22 (24)	10 (30.3)	12 (21.3)	0.32
Fever	42 (46)	15 (45.5)	27 (47.3)	0.86
CT bilateral patchy shadow	41 (45.5)	23 (69.5)	18 (32.3)	0.005
CT pleural effusion	27 (30)	4 (13)	23 (40.4)	0.024
CT pulmonary consolidations	42 (47)	14 (43.3)	28 (48.6)	0.69
Median PaO_2_/FiO_2_ baseline	297 (109)	261 (138)	304 (106)	0.06
Median PaO_2_/Fio_2_ nadir	215 (201)	138 (128)	252 (191)	0.017
White blood cells count/mm^3^	7827 (3626)	9254 (2561)	6992 (3915)	0.014
Baseline Lymphocytes /mm^3^	1258 (2364)	851 (579)	1477 (2890)	0.32
Baseline C-reactive protein (mg/dL)	8.1 (6.3)	11.3 (7)	6.1 (5.1)	0.002
In-hospital death or ICU admission (%)	22 (24)	9 (27.1)	13 (22.8)	0.63

Data are expressed as mean and standard deviation, median (interquartile range) and number (%) as appropriate. Significant *p* values are marked in bold. CT: computed tomography; ECT: early corticosteroid therapy; COPD: chronic obstructive pulmonary disease. ICU: intensive care unit.

**Table 2 geriatrics-07-00021-t002:** List of oral corticosteroids.

Corticosteroid (CCS) Home Therapy	Observed Frequency (%)	Total CCS Given Dose in mg (*n*° of Subjects)
Dexamethasone 4 mg tabs	3/33 (9.1)	50–100 (3)
Methylprednisolone sodium succinate 16 mg	29/33 (87.9)	50–100 (6) 100–300 (1) >300 (9)
Prednisone 25 mg po tabs	1/33 (3.0)	50–100 (3)

## Data Availability

The datasets used and/or analysed during the current study are available from the corresponding author on reasonable request.

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
