# Peer review of "Pitfalls of Early Systemic Corticosteroids Home Therapy in Older Patients with COVID-19 Pneumonia"

_geriatrics, 2022, doi:10.3390/geriatrics7010021_

Round 1

Reviewer 1 Report

The study emphasizes the importance of early therapy with corticosteroid drugs on the clinical evolution of Covid 19 infection.

The patients are elderly and therefore immunologically compromised.

The manuscript deserves special consideration, however it is suggested to evaluate this publication to better define the concepts of the conclusions:

Observational Study PLoS One

. 2022 Jan 21;17(1):e0261711. doi: 10.1371/journal.pone.0261711. eCollection 2022.

Use of glucocorticoids megadoses in SARS-CoV-2 infection in a spanish registry: SEMI-COVID-19

Author Response

REVIEWER #1

COMMENT 1:

The study emphasizes the importance of early therapy with corticosteroid drugs on the clinical evolution of Covid 19 infection. The patients are elderly and therefore immunologically compromised. The manuscript deserves special consideration, however it is suggested to evaluate this publication to better define the concepts of the conclusions:

 Observational Study PLoS One . 2022 Jan 21;17(1):e0261711. doi: 10.1371/journal.pone.0261711. eCollection 2022. Use of glucocorticoids megadoses in SARS-CoV-2 infection in a spanish registry: SEMI-COVID-19 ­>> line 145

ANSWER 1: We thank the Reviewer for the suggestion. We have added the reference and revised the Discussion section accordingly. Please see lines 149-151.

Reviewer 2 Report

Dear authors, you have dine good manuscript, i have few comments 

it will be great if you consider them . 

Abstract

Line 11 and 12- After the aim it would help to mention here in a sentence of two the value or what informed the study, what it will achieve

Introduction

Line 32 and 33- have a reference for the three stages you have listed in this part Line 49- Elaborate further in a few lines why this study was important, what is it that is missing in the current literature that it will address and deliver on

Conclusion

Line 197-198 you mention older patient, does it mean they showed benefit for the younger people? Please clarify this ion the sentence.

Author Response

REVIEWER #2

COMMENT 1:

Dear authors, you have done good manuscript, i have few comments 

it will be great if you consider them. 

Abstract

  • Line 11 and 12- After the aim it would help to mention here in a sentence of two the value or what informed the study, what it will achieve

ANSWER 1

We are grateful to the reviewer for the comment, we have better explained the value and the clinical implications of the current study in the Introduction Section (Please see lines 46-49), in order to respect the Geriatrics Journal guidelines for word count on the abstract.

COMMENT 2:

Introduction

  • Line 32 and 33- have a reference for the three stages you have listed in this part 

ANSWER 2:

We thank the Reviewer for the suggestion, we have included the reference listing the three stages of the disease.

COMMENT 3:

  • Line 49- Elaborate further in a few lines why this study was important, what is it that is missing in the current literature that it will address and deliver on

ANSWER 3:

We thank the Reviewer for giving us the chance to better explain this point. Accordingly, we have added the strength points of the study and the possible implications in the Introduction section (Please see Page 2, lines 46-49).

 COMMENT 4:

Conclusion

  • Line 197-198 you mention older patient, does it mean they showed benefit for the younger people? Please clarify this ion the sentence.

ANSWER 4:

We are grateful to the Reviewer for pointing out this issue; our study confirmed a detrimental effect of an untimely assumption of corticosteroids in older patients, as previously reported in younger cohorts of patients. We modified the conclusion section accordingly to better explain to the reader this important concept (Please see Page 6, lines 204-205)
